# Modulated Phase Diffusor: Content-Oriented Feature Synthesis for Detecting Unknown Objects

**Aming Wu,   Cheng Deng**[*]
School of Electronic Engineering, Xidian University, Xi'an, China
`amwu@xidian.edu.cn, chdeng@mail.xidian.edu.cn`

## Abstract

To promote the safe deployment of object detectors, a task of unsupervised out-of-distribution object detection (OOD-OD) is recently proposed, aiming to detect unknown objects during training without reliance on any auxiliary OOD data. To alleviate the impact of lacking OOD data, for this task, one feasible solution is to exploit the known in-distribution (ID) data to synthesize proper OOD information for supervision, which strengthens detectors' discrimination. From the frequency perspective, since the phase generally reflects the content of the input, in this paper, we explore leveraging the phase of ID features to generate expected OOD features involving different content. And a method of Modulated Phase Diffusion (MPD) is proposed, containing a shared forward and two different reverse processes. Specifically, after calculating the phase of the extracted features, to prevent the rapid loss of content in the phase, the forward process gradually performs Gaussian Average on the phase instead of adding noise. The averaged phase and original amplitude are combined to obtain the features taken as the input of the reverse process. Next, one OOD branch is defined to synthesize virtual OOD features by continually enlarging the content discrepancy between the OOD features and original ones. Meanwhile, another modulated branch is designed to generate augmented features owning a similar phase as the original features by scaling and shifting the OOD branch. Both original and augmented features are used for training, enhancing the discrimination. Experimental results on OOD-OD, incremental object detection, and open-set object detection demonstrate the superiorities of our method. The source code will be released at https://github.com/AmingWu/MPD.

## 1 Introduction

Detecting unknown objects is critical for the safe application of detection systems. Currently, most detection methods (Ren et al., 2015; He et al., 2017; Carion et al., 2020) usually follow a closed-set assumption, i.e., the training and testing processes share the same category space. However, the real scenario is open and filled with unknown objects, presenting enormous challenges for closed-set assumption based detectors. To facilitate the deployment of object detectors, a task of unsupervised out-of-distribution object detection (OOD-OD) (Du et al., 2022c) is recently proposed, whose purpose is to detect unknown OOD objects during training without exploiting any auxiliary OOD data.

Due to lacking OOD data for training, the challenge of unsupervised OOD-OD (Du et al., 2022c) mainly lies in how to only leverage the known in-distribution (ID) data to enhance the ability of distinguishing OOD objects while reducing the impact on the performance of detecting ID objects. One feasible solution (Du et al., 2022c; Reiss et al., 2022) is to synthesize a series of proper virtual OOD features for supervision based on the ID data, which is conducive to promoting the object detector to learn a clear boundary between ID and OOD objects. Particularly, the work (Du et al., 2022c) first leverages ID data to estimate class-conditional distribution for each category. Then, virtual OOD features are sampled from the region that slightly deviates from the estimated distribution. However, to estimate the distributions accurately, it is important to utilize abundant objects for each

---

[*]Corresponding Author

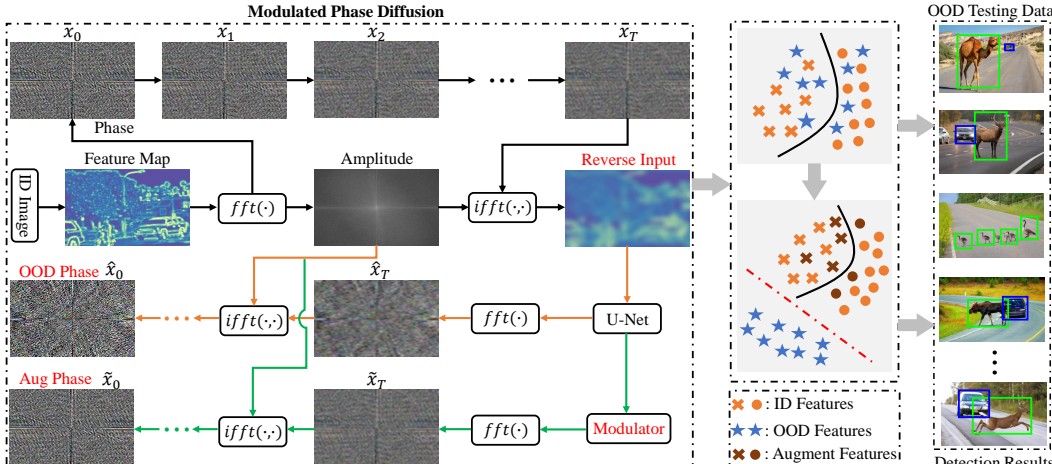

Figure 1: Modulated Phase Diffusion for OOD-OD, which exploits the Phase information for content-oriented feature synthesis and consists of a shared forward and two different reverse processes. To alleviate the impact of lacking unknown data, the first reverse branch (as shown in orange arrows) is to synthesize virtual OOD features that differ from the content of the ID features. Meanwhile, another branch (as shown in green lines) aims to synthesize the augmented features of the ID features. By taking these synthesized features for training, our method could enhance the ability of discriminating OOD objects while reducing the impact on detecting ID objects (as shown in red and black lines).

category, which limits its application in the case of few samples. Meanwhile, when the number of categories is large, estimating the distribution for each category may lead to an increase of computational costs.

In this paper, we still explore feature synthesis to alleviate the impact of lacking OOD data. In general, the extracted features of input images could be recognized as containing style and content information (Lee et al., 2023). And the content of the OOD features should be different from that of the ID features. Besides, from the frequency perspective, recent researches (Lee et al., 2023; Chen et al., 2021; Oppenheim & Lim, 1981) have shown that the amplitude and phase could be separately regarded as the style and content of the input. To this end, we pay more attention to exploiting the phase information to perform content-oriented feature synthesis, which is instrumental in addressing the key challenge of unsupervised OOD-OD (Du et al., 2022c).

Specifically, as shown in Fig. 1, an approach of Modulated Phase Diffusion (MPD) is proposed, which is a dedicated phase diffusion generator and mainly consists of a shared forward and two different reverse processes. After extracting the features of the input images, we first utilize Fourier transform to decompose the corresponding amplitude and phase components. For traditional diffusion models (Ho et al., 2020; Luo, 2022; Rombach et al., 2022), the forward process is to gradually add noise to progressively weaken the content of the input. However, experimental results show that directly adding noise into the phase could not boost the performance of discriminating OOD objects. The reason may be that the phase components generally describe sensitive angle-related information (Oppenheim & Lim, 1981). Adding much noise may rapidly destroy the content in the phase. Therefore, during the forward process, Gaussian Average is gradually performed on the phase, alleviating the loss speed of content in phase. The averaged phase and original amplitude are input into the inverse Fourier transform, whose output is taken as the input of the reverse process. Next, by continually enlarging the content discrepancy between the OOD features and ID features, one OOD branch (as shown in orange lines) is designed to synthesize expected virtual OOD features. Meanwhile, another modulated branch (as shown in green lines) is presented to generate augmented features owning a similar phase as the ID features by scaling and shifting the OOD branch. Finally, these synthesized features are used for training, enhancing the discrimination ability. Experimental results on multiple datasets demonstrate the superiorities of our method.

In summary, our contributions are mainly three-fold: (1) To alleviate the impact of lacking unknown data, we explore leveraging the phase information to perform content-oriented feature synthesis, which is conducive to improving the discrimination. (2) To generate expected features, a method of Modulated Phase Diffusion is proposed to synthesize virtual OOD features and augmented ID features, which is instrumental in addressing the key challenge of unsupervised OOD-OD. (3) In the

experiments, our method is evaluated on OOD-OD (Du et al., 2022c), incremental object detection (Kj et al., 2021), and open-set object detection (Han et al., 2022). Particularly, for OpenImages dataset (Kuznetsova et al., 2020), compared with the baseline method (Du et al., 2022c), our method significantly reduces FPR95 by around **8.78%**.

## 2 RELATED WORK

**OOD Detection.** Discovering unknown objects is an important ability of human intelligence. To simulate this ability, OOD detection (Hendrycks & Gimpel, 2017; Liang et al., 2017; Wu & Deng, 2023a;b) has recently attracted much attention, whose goal is to discriminate OOD data from ID data. Currently, most methods (Geifman & El-Yaniv, 2019; Jeong & Kim, 2020; Katz-Samuels et al., 2022; Malinin & Gales, 2018; Meinke & Hein, 2020) focus on OOD image classification and try to design regularization methods to improve the discrimination. Particularly, many methods (Hendrycks et al., 2019; Liu et al., 2020; Bendale & Boult, 2016; Wang et al., 2021; Sun et al., 2022) aim to design some specific score mechanisms to regularize the model to produce different scores for OOD and ID data. And the score could be used to distinguish OOD data from ID data. Besides, reconstruction (Denouden et al., 2018; Zhou, 2022) is also a commonly used idea for OOD detection, which assumes that the reconstruction loss for OOD data is usually larger than that for ID data. Though these methods have been shown to be effective, since object detection involves object localization and classification, these methods could not be directly applied to OOD-OD.

Recently, a task of unsupervised OOD-OD (Du et al., 2022c; Wu et al., 2023) is proposed to localize and recognize unknown objects during training. To reduce the impact of lacking OOD data for supervision, Du et al. (Du et al., 2022c) proposed to use large-scale object samples to estimate the distribution of each category, which is used to sample virtual OOD features that slightly deviate from the estimated distribution. However, when the number of categories is large, estimating the distribution for each category may increase computational costs. Besides, the work (Du et al., 2022b) presented to learn unknown-aware knowledge from auxiliary videos, which does not match the setting of unsupervised OOD-OD. Finally, Du et al. (Du et al., 2022a) further presented to use a distance-based mechanism to shape the learned representations. Different from the above methods, in this paper, we explore leveraging the phase components to perform content-oriented feature synthesis, which is beneficial for improving the discrimination ability of the object detector.

**Diffusion Models.** As a popular generator, diffusion models (Rombach et al., 2022; Yang et al., 2022b) generally contain forward diffusion for gradually adding noise and a reverse process to continually recover the denoised data. Particularly, Ho et al. (Ho et al., 2020) first propose Denoising Diffusion Probabilistic Models, accelerating the deployment of diffusion models. Based on this work, some methods (Rombach et al., 2022; Gu et al., 2022; Hu et al., 2022) explore introducing the attention mechanism (Vaswani et al., 2017) and Variational AutoEncoder (VAE) (Van Den Oord et al., 2017) into existing diffusion models, which produce stable diffusion models and generate high-quality images. However, diffusion models are rarely used for specific feature synthesis. In this paper, we propose a method of Modulated Phase Diffusion, which is a dedicated phase-based diffusion generator including a shared forward and two different reverse processes. By this diffusion model, we can obtain expected OOD features and augmented ID features, which strengthens the discrimination. Experimental results on multiple datasets demonstrate the superiorities of our method.

## 3 MODULATED PHASE DIFFUSION FOR FEATURE SYNTHESIS

For unsupervised OOD-OD (Du et al., 2022c), the key is how to overcome the difficulty of lacking OOD data. To this end, we design a Modulated Phase Diffusion for synthesizing specific features. Concretely, we follow the settings (Du et al., 2022c) and only utilize the ID data that own a fixed number of categories for training. During inference, the object detector should distinguish ID objects from OOD objects accurately.

### 3.1 FORWARD DIFFUSION VIA AVERAGING PHASE

As shown in Fig. 2, we follow the baseline work (Du et al., 2022c) and exploit the widely used object detector, i.e., Faster R-CNN (Ren et al., 2015; He et al., 2017), as the basic detection model.

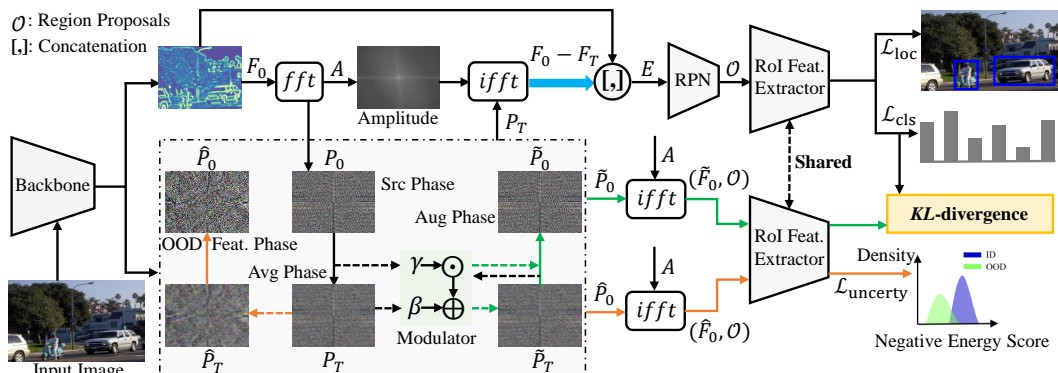

Figure 2: The details of MPD for detecting unknown objects. MPD mainly contains three functions: (1) The output of the forward process is used to enhance object-related information in the features extracted by the backbone network, which is conducive to improving localization performance. (2) To alleviate the impact of lacking OOD data, one OOD branch (as shown in orange lines) is presented to synthesize expected OOD features $\hat{F}_0$ that differ from the content of the original features $F_0$. (3) Another modulated branch (as shown in green lines) is designed to generate augmented features $\tilde{F}_0$ of the original features $F_0$, strengthening the discrimination of the object classifier.

Given an input image, a backbone network, e.g., ResNet (He et al., 2016), is utilized to extract the corresponding feature map $F_0 \in \mathbb{R}^{w \times h \times c}$, where $w$, $h$, and $c$ separately denote width, height, and the number of channels.

Then, $F_0$ is input into fast Fourier transform (FFT) (Cochran et al., 1967; Lee et al., 2023) to decompose the corresponding amplitude $A \in \mathbb{R}^{w \times h \times c}$ and phase $P_0 \in \mathbb{R}^{w \times h \times c}$. Next, to attain content-oriented feature synthesis, the diffusion operation is performed on the phase $P_0$ containing the content information of the input. In general, the forward diffusion (Yang et al., 2022b; Ho et al., 2020) gradually adds noise to progressively cover up the content of the input image. Unfortunately, experimental results show that adding noise into the phase of input features could not boost the performance. Taking PASCAL VOC (Everingham et al., 2010) and MS-COCO (Lin et al., 2014) as the ID data for training and OOD data for evaluation, based on the FPR95 metric, the performance of using the traditional diffusion model (Ho et al., 2020) and adding noise into the phase is increased by **6.48%**. The reason may be that the phase of the features describes sensitive angle-related information (Oppenheim & Lim, 1981). Adding much noise may rapidly destroy the content in the phase, affecting the performance of the diffusion model.

To this end, we explore replacing adding noise with Gaussian Average, which is beneficial for slowing down the loss speed of content in phase. Specifically, the forward process is fixed to a Markov chain that gradually performs Gaussian Average on the phase $P_0$:

$$P_t = P_{t-1} * \mathcal{G}(\sigma), \quad \mathcal{G}(\sigma) = \frac{1}{2\pi\sigma^2} e^{-(i^2+j^2)/2\sigma^2}, \tag{1}$$

where $\mathcal{G}(\sigma)$ represents Gaussian Kernel with the variance $\sigma$. $(i, j)$ indicates the position in the kernel. $t = 1, ..., T$. $P_t \in \mathbb{R}^{w \times h \times c}$ is the convolutional output. Meanwhile, the kernel size is set to $5 \times 5$. Finally, it is worth noting that the forward process has no learnable parameters.

Since Gaussian Average is a local operation, compared with globally adding noise on the phase, this operation reduces the damage to the content involved in $P_0$. Finally, after multiple iterations, the averaged phase $P_T \in \mathbb{R}^{w \times h \times c}$ and original amplitude $A$ are input to inverse fast Fourier transform (IFFT) (Cochran et al., 1967; Lee et al., 2023) to acquire the output $F_T \in \mathbb{R}^{w \times h \times c}$ that is taken as the input of the following reverse stage to synthesize specific features.

## 3.2 REVERSE PROCESS FOR OOD FEATURE SYNTHESIS

To reduce the impact of lacking OOD data, as shown in Fig. 2, one OOD reverse branch is presented to use $F_T$ to synthesize expected OOD features that differ from the content of $F_0$, which is instrumental in improving the ability of discriminating OOD objects from ID objects. Concretely, in Fig. 1 and 2, taking $F_T$ as the input, a U-Net model (Ronneberger et al., 2015) is designed to predict the

feature $\hat{D}_{t-1} \in \mathbb{R}^{w \times h \times c}$. Then, we perform FFT on $\hat{D}_{t-1}$ to decompose the corresponding phase $\hat{P}_{t-1} \in \mathbb{R}^{w \times h \times c}$. Next, $\hat{P}_{t-1}$ and $A$ are input to IFFT, whose output is taken as the input of the next step. The processes are as follows:

$$\hat{D}_{t-1} = \epsilon_\theta(\hat{F}_t, \ t), \quad \hat{P}_{t-1} = fft(\hat{D}_{t-1}), \quad \hat{F}_{t-1} = ifft(\hat{P}_{t-1}, \ A), \tag{2}$$

where $\epsilon_\theta(\cdot, \cdot)$ represents the learned U-Net model. And $t = T, ..., 1$. $\hat{F}_T = F_T$. Since the amplitude generally describes the style information, to alleviate the style impact and focus on content-oriented feature synthesis, we replace the amplitude from $\hat{D}_{t-1}$ with the original amplitude $A$. $\hat{F}_{t-1} \in \mathbb{R}^{w \times h \times c}$ is the output at the timestep $t$. Next, taking $\hat{F}_{t-1}$ as the input, we continually perform the above operations to obtain the synthesized virtual OOD map $\hat{F}_0 \in \mathbb{R}^{w \times h \times c}$.

During training, a loss function $\mathcal{L}_{ood}$ is proposed to facilitate the designed U-Net model to possess the capability of OOD feature synthesis:

$$\mathcal{L}_{ood} = \mathbb{E}_t[||\epsilon_t - fft(\epsilon_\theta(\hat{F}_t, \ t))||^2], \quad \epsilon_t = P_{T-t} - P_{T-t+1}, \tag{3}$$

where $\epsilon_t$ describes the lost phase information in $P_0$ from the timestep $T - t$ to $T - t + 1$. Since the average operation is continually performed on $P_0$, as the iteration increases, the information about $P_0$ involved in $\epsilon_t$ gradually decreases, which is conducive to promoting $\hat{F}_0$ to contain rich information that differs from the content in $F_0$. Finally, to further enlarge the gap between $\hat{F}_0$ and $F_0$, a loss $\mathcal{L}_{\text{dis}}$ is defined to maximize the $KL$-divergence between the virtual OOD features and ID features, i.e., $\mathcal{L}_{\text{dis}} = \text{KL}[q(\hat{F}_0), \ q(F_0)]$, where $q(\cdot)$ represents the probability distribution.

## 3.3 MODULATED REVERSE PROCESS FOR AUGMENTED FEATURE SYNTHESIS

The modulated reverse process aims to recover the original features from the forward output $F_T$, which is taken as the augmentation for training and is instrumental in strengthening the discrimination of the classifier for ID objects. Specifically, as shown in Fig. 1 and 2, taking $F_T$ as the input, we still employ the U-Net model of the OOD branch to predict the corresponding feature $\tilde{D}_{t-1} \in \mathbb{R}^{w \times h \times c}$. Since $\tilde{D}_{t-1}$ involves plentiful OOD-related information, we exploit the modulation mechanism (Perez et al., 2018; Wang et al., 2020) to perform a transformation on $\tilde{D}_{t-1}$:

$$\tilde{D}_{t-1} = \epsilon_\theta(\tilde{F}_t, \ t), \quad \mathcal{D}_{t-1} = \gamma \odot \tilde{D}_{t-1} + \beta, \tag{4}$$

where $\gamma$ and $\beta$ are learnable parameters for channel-wisely scaling and shifting $\tilde{D}_{t-1}$. $\mathcal{D}_{t-1} \in \mathbb{R}^{w \times h \times c}$ and $t = T, ..., 1$. $\tilde{F}_T = F_T$. Compared with employing a newly-designed U-Net model, leveraging the modulation mechanism (Perez et al., 2018; Wang et al., 2020) does not introduce a large number of parameters, alleviating the overfitting risk. Then, FFT is performed on $\mathcal{D}_{t-1}$ to decompose the phase $\mathcal{P}_{t-1} \in \mathbb{R}^{w \times h \times c}$, which corresponds to the lost phase caused by the forward average. Next, we utilize the IFFT operation to calculate the input of the next iteration:

$$\mathcal{P}_{t-1} = fft(\mathcal{D}_{t-1}), \quad \tilde{P}_{t-1} = \tilde{P}_t + \mathcal{P}_{t-1}, \quad \tilde{F}_{t-1} = ifft(\tilde{P}_{t-1}, \ A), \tag{5}$$

where $\tilde{P}_T = P_T$. $\tilde{P}_{t-1} \in \mathbb{R}^{w \times h \times c}$ represents the recovered phase. Similarly, to reduce the impact of amplitude, $\tilde{P}_{t-1}$ and original amplitude $A$ are input to IFFT to acquire the output $\tilde{F}_{t-1} \in \mathbb{R}^{w \times h \times c}$. Next, we continually perform the above operations in equation 4 and equation 5 to generate the augmented feature map $\tilde{F}_0 \in \mathbb{R}^{w \times h \times c}$ of the original feature map $F_0$.

During training, a loss $\mathcal{L}_{aug}$ is defined to promote the modulated parameters to contain the ability of transforming OOD information:

$$\mathcal{L}_{aug} = \mathbb{E}_t[||\epsilon_t - fft(\mathcal{D}_{t-1})||^2], \quad \epsilon_t = P_{t-1} - P_t, \tag{6}$$

where $\epsilon_t$ describes the lost phase information from the timestep $t - 1$ to $t$. By optimizing equation 6, the decomposed phase $\mathcal{P}_{t-1}$ could be facilitated to contain rich information about the lost phase, which makes $\tilde{F}_0$ retain plentiful content about $F_0$.

## 3.4 MPD-DRIVEN OOD OBJECT DETECTION

In general, object detection involves two subtasks, i.e., object localization and recognition. Therefore, enhancing object-related information in $F_0$ is instrumental in detecting objects accurately.

---

**Algorithm 1** Modulated Phase Diffusion for Unsupervised OOD-OD

---

**Input:** ID data $\{X, Y\}$, randomly initialized detector with parameter $\varphi$, randomly initialized U-Net with parameter $\theta$, randomly initialized modulator with parameter $\gamma$ and $\beta$, weight $\lambda$ for the $KL$-loss, weight $\alpha$ for the loss $\mathcal{L}_{MPD}$, weight $\tau$ for the uncertainty loss $\mathcal{L}_{\text{uncerty}}$.
**Output:** Object detector with parameter $\varphi^*$, and OOD detector $\mathcal{C}$.
**while** *train* **do**

    Sample images from the ID dataset $\{X, Y\}$.
    Perform the forward phase diffusion using equation 1 to obtain $P_T$ and $F_T$.
    **for** *t = T, ..., 1* **do**
        $\hat{D}_{t-1} = \epsilon_\theta(\hat{F}_t, t)$, $\hat{P}_{t-1} = fft(\hat{D}_{t-1})$, and $\hat{F}_{t-1} = ifft(\hat{P}_{t-1}, A)$.    # OOD map $\hat{F}_0$
        $\tilde{D}_{t-1} = \epsilon_\theta(\tilde{F}_t, t)$, and $\mathcal{D}_{t-1} = \gamma \odot \tilde{D}_{t-1} + \beta$.
        $\mathcal{P}_{t-1} = fft(\mathcal{D}_{t-1})$, $\tilde{P}_{t-1} = \tilde{P}_t + \mathcal{P}_{t-1}$, and $\tilde{F}_{t-1} = ifft(\tilde{P}_{t-1}, A)$.    # Aug map $\tilde{F}_0$
    **end**
    Calculate the overall training objective $\mathcal{L}$ using equation 3, equation 6, equation 7, equation 8, and equation 9.
    Update the parameters $\varphi$, $\theta$, $\gamma$, and $\beta$ based on equation 9.
**end**
**while** *eval* **do**
    Calculate the OOD uncertainty score using the left part of equation 10.
    Perform thresholding comparison using the right part of equation 10.
**end**

---

To this end, as shown in Fig. 2, a residual operation between $F_0$ and $F_T$ is first performed. Since the content in the forward output $F_T$ is blurry after multiple average operations, the residual result contains plentiful object-related information. Then, the residual result is concatenated with $F_0$ to obtain the enhanced result $E \in \mathbb{R}^{w \times h \times c}$, i.e., $E = \Psi([F_0, F_0 - F_T])$, where $\Psi(\cdot) \in \mathbb{R}^{1 \times 1 \times 2c \times c}$ represents one-layer convolution to transform the number of channels.

Next, $E$ is taken as the input of the RPN module (Ren et al., 2015; He et al., 2017) to output a set of object proposals $\mathcal{O}$. Meanwhile, based on $\mathcal{O}$, RoI-Alignment followed by RoI-Feature extraction (He et al., 2017) is separately performed on $E$ and augmented map $\tilde{F}_0$ to obtain $O_{\text{in}} \in \mathbb{R}^{m \times n}$ and $O_{\text{aug}} \in \mathbb{R}^{m \times n}$, where $m$ and $n$ denote the number of proposals and channels. Then, $O_{\text{in}}$ is input to the object classifier and regressor to calculate the classification loss $\mathcal{L}_{\text{cls}}$ and localization loss $\mathcal{L}_{\text{loc}}$:

$$\mathcal{L}_{\text{in}} = \mathcal{L}_{\text{cls}} + \mathcal{L}_{\text{loc}} + \lambda \cdot \text{KL}[p(O_{\text{in}}), \; p(O_{\text{aug}})], \tag{7}$$

where $\lambda$ is a hyper-parameter, which is set to 0.001 in the experiments. The $KL$-divergence loss is to constrain the prediction consistency between $O_{\text{in}}$ and $O_{\text{aug}}$, ameliorating the ability of the object classifier for discriminating ID objects.

Finally, to distinguish OOD objects from ID objects, based on $\mathcal{O}$, RoI-Alignment followed by RoI-Feature extraction is performed on $\hat{F}_0$ to extract OOD features $O_{\text{ood}} \in \mathbb{R}^{m \times n}$. $O_{\text{ood}}$ and $O_{\text{in}}$ are used to compute an uncertainty loss (Du et al., 2022c), regularizing the detector to produce a low OOD score for the ID object features, and a high OOD score for the virtual OOD features:

$$\mathcal{L}_{\text{uncerty}} = \mathbb{E}_{u \backsim O_{\text{in}}}[-\log\frac{\exp^{-\mathbf{E}(u)}}{1 + \exp^{-\mathbf{E}(u)}}] + \mathbb{E}_{v \backsim O_{\text{ood}}}[-\log\frac{1}{1 + \exp^{-\mathbf{E}(v)}}], \tag{8}$$

where $\mathbf{E}(\cdot)$ is the object-level energy score (Du et al., 2022c; Liu et al., 2020). **The overall objective** is shown as follows:

$$\mathcal{L} = \mathcal{L}_{\text{in}} + \alpha \cdot \mathcal{L}_{MPD} + \tau \cdot \mathcal{L}_{\text{uncerty}}, \quad \mathcal{L}_{MPD} = \mathcal{L}_{ood} + \mathcal{L}_{aug} - \mathcal{L}_{\text{dis}}, \tag{9}$$

where $\alpha$ and $\tau$ are two hyper-parameters, which are set to 0.001 and 0.1 in the experiments.

### 3.5 INFERENCE FOR OOD OBJECT DETECTION

During inference, we only leverage the forward phase diffusion to acquire the feature $F_T$ used to enhance object-related information. Meanwhile, we only calculate the uncertainty score for OOD

Table 1: The performance (%) of unsupervised OOD-OD. ↑ denotes larger values are better and ↓ represents smaller values are better.

| In-distribution Data | Method | FPR95 ↓ | AUROC ↑ | mAP (ID)↑ |
|---|---|---|---|---|
| | | OOD: MS-COCO / OpenImages | | |
| PASCAL-VOC | MSP (Hendrycks & Gimpel, 2017) | 70.99 / 73.13 | 83.45 / 81.91 | 48.7 |
| | ODIN (Liang et al., 2017) | 59.82 / 63.14 | 82.20 / 82.59 | 48.7 |
| | Mahalanobis (Lee et al., 2018b) | 67.73 / 65.41 | 81.45 / 81.48 | 48.7 |
| | Gram matrices (Sastry & Oore, 2020) | 62.75 / 67.42 | 79.88 / 77.62 | 48.7 |
| | Energy score (Liu et al., 2020) | 56.89 / 58.69 | 83.69 / 82.98 | 48.7 |
| | Generalized ODIN (Hsu et al., 2020) | 59.57 / 70.28 | 83.12 / 79.23 | 48.1 |
| | CSI (Tack et al., 2020) | 59.91 / 57.41 | 81.83 / 82.95 | 48.1 |
| | GAN-synthesis (Lee et al., 2018a) | 60.93 / 59.97 | 83.67 / 82.67 | 48.5 |
| | SIREN-vMF (Du et al., 2022a) | 64.68 / 68.53 | 85.36 / 82.78 | - |
| | SIREN-KNN (Du et al., 2022a) | 47.45 / 50.38 | 89.67 / **88.80** | - |
| | VOS (Baseline) (Du et al., 2022c) | 47.53 / 51.33 | 88.70 / 85.23 | 48.9 |
| | **MPD** | **41.28 / 46.45** | **90.54** / 88.03 | **49.2** |
| Berkeley DeepDrive-100k | MSP (Hendrycks & Gimpel, 2017) | 80.94 / 79.04 | 75.87 / 77.38 | 31.2 |
| | ODIN (Liang et al., 2017) | 62.85 / 58.92 | 74.44 / 76.61 | 31.2 |
| | Mahalanobis (Lee et al., 2018b) | 55.74 / 47.69 | 85.71 / 88.05 | 31.2 |
| | Gram matrices (Sastry & Oore, 2020) | 60.93 / 77.55 | 74.93 / 59.38 | 31.2 |
| | Energy score (Liu et al., 2020) | 60.06 / 54.97 | 77.48 / 79.60 | 31.2 |
| | Generalized ODIN (Hsu et al., 2020) | 57.27 / 50.17 | 85.22 / 87.18 | **31.8** |
| | CSI (Tack et al., 2020) | 47.10 / 37.06 | 84.09 / 87.99 | 30.6 |
| | GAN-synthesis (Lee et al., 2018a) | 57.03 / 50.61 | 78.82 / 81.25 | 31.4 |
| | VOS (Baseline) (Du et al., 2022c) | 44.27 / 35.54 | 86.87 / 88.52 | 31.3 |
| | **MPD** | **37.24 / 26.76** | **88.56 / 92.23** | 31.4 |

Figure 3: Results on the OOD images from MS-COCO. The first and second rows respectively indicate results based on VOS (Du et al., 2022c) and our method. The ID dataset is BDD-100k.

object detection (Du et al., 2022c). Specifically, for a predicted bounding box **b**, the processes of distinguishing OOD objects are shown as follows:

$$\mathcal{S} = \frac{\exp^{-\mathbf{E}(\mathbf{b})}}{1 + \exp^{-\mathbf{E}(\mathbf{b})}}, \qquad \mathcal{C}(\mathbf{b}) = \begin{cases} 0 & \text{if } \mathcal{S} < \delta, \\ 1 & \text{if } \mathcal{S} \geq \delta. \end{cases} \tag{10}$$

For the output of the classifier $\mathcal{C}(\cdot)$, we use the threshold mechanism (Du et al., 2022c) to distinguish ID objects (the result is 1) from OOD objects (the result is 0). The threshold $\delta$ is commonly set to 0.95 so that a high fraction of ID data is correctly classified. Finally, Algorithm 1 shows the training and testing processes of our method.

## 4 EXPERIMENTS

For unsupervised OOD-OD, our method is first evaluated on two different benchmarks (Du et al., 2022c). Then, to further demonstrate the effectiveness, we evaluate our method on class-incremental object detection (IOD) (Kj et al., 2021) and open-set object detection (OSOD) (Han et al., 2022).

### 4.1 OOD-OD PERFORMANCE ANALYSIS

Table 1 shows the performance of unsupervised OOD-OD. We can see that though different methods own a similar mAP performance, the ability of detecting OOD objects differs significantly. This

Table 2: Performance (%) analysis of class-incremental object detection. 'iOD + Ours' indicates that our method is plugged into iOD (Kj et al., 2021). Here, '50' and '75' separately represent that the mAP metric is calculated when the IOU threshold is set to 0.5 and 0.75.

| 10 + 10 setting | aero | cycle | bird | boat | bottle | bus | car | cat | chair | cow | table | dog | horse | bike | person | plant | sheep | sofa | train | tv | mAP |
|---|---|---|---|---|---|---|---|---|---|---|---|---|---|---|---|---|---|---|---|---|---|
| OW-DETR (50) (Gupta et al., 2022) | 61.8 | 69.1 | 67.8 | 45.8 | 47.3 | 78.3 | 78.4 | 78.6 | 36.2 | 71.5 | 57.5 | 75.3 | 76.2 | 77.4 | 79.5 | 40.1 | 66.8 | 66.3 | 75.6 | 64.1 | 65.7 |
| ROSETTA (50) (Yang et al., 2022a) | 74.2 | 76.2 | 64.9 | 54.4 | 57.4 | 76.1 | 84.4 | 68.8 | 52.4 | 67.0 | 62.9 | 63.3 | 79.8 | 72.8 | 78.1 | 40.1 | 62.3 | 61.2 | 72.4 | 66.8 | 66.8 |
| iOD (50) (Kj et al., 2021) | 76.0 | 74.6 | 67.5 | 55.9 | 57.6 | 75.1 | 85.4 | 77.0 | 43.7 | 70.8 | 60.1 | 66.4 | 76.0 | 72.6 | 74.6 | 39.7 | 64.0 | 60.2 | 68.5 | 60.5 | 66.3 |
| iOD + Ours (50) | 76.0 | 75.7 | 70.0 | 52.5 | 55.3 | 78.9 | 85.2 | 76.5 | 45.5 | 75.3 | 57.2 | 79.5 | 79.8 | 76.4 | 79.8 | 43.5 | 71.7 | 69.0 | 74.2 | 68.0 | **69.5** |
| iOD (75) (Kj et al., 2021) | 39.0 | 36.5 | 28.4 | 19.4 | 24.2 | 47.2 | 56.7 | 41.0 | 19.1 | 48.0 | 21.1 | 32.1 | 43.0 | 36.3 | 40.0 | 14.8 | 40.1 | 36.5 | 37.3 | 45.3 | 35.3 |
| iOD + Ours (75) | 42.2 | 41.3 | 29.6 | 22.3 | 22.8 | 53.7 | 58.0 | 41.4 | 21.5 | 42.6 | 24.6 | 32.9 | 39.8 | 41.4 | 38.5 | 15.5 | 44.2 | 36.5 | 35.0 | 45.6 | **36.5** |

| 15 + 5 setting | aero | cycle | bird | boat | bottle | bus | car | cat | chair | cow | table | dog | horse | bike | person | plant | sheep | sofa | train | tv | mAP |
|---|---|---|---|---|---|---|---|---|---|---|---|---|---|---|---|---|---|---|---|---|---|
| OW-DETR (50) (Gupta et al., 2022) | 77.1 | 76.5 | 69.2 | 51.3 | 61.3 | 79.8 | 84.2 | 81.0 | 49.7 | 79.6 | 58.1 | 79.0 | 83.1 | 67.8 | 85.4 | 33.2 | 65.1 | 62.0 | 73.9 | 65.0 | 69.4 |
| ROSETTA (50) (Yang et al., 2022a) | 76.5 | 77.5 | 65.1 | 56.0 | 60.0 | 78.3 | 85.5 | 78.7 | 49.5 | 68.2 | 67.4 | 71.2 | 83.9 | 75.7 | 82.0 | 43.0 | 60.6 | 64.1 | 72.8 | 67.4 | 69.2 |
| iOD (50) (Kj et al., 2021) | 78.4 | 79.7 | 66.9 | 54.8 | 56.2 | 77.7 | 84.6 | 79.1 | 47.7 | 75.0 | 61.8 | 74.7 | 81.6 | 77.5 | 80.2 | 37.8 | 58.0 | 54.6 | 73.0 | 56.1 | 67.8 |
| iOD + Ours (50) | 77.5 | 78.7 | 71.7 | 54.7 | 62.3 | 78.7 | 84.5 | 77.0 | 51.3 | 78.8 | 66.1 | 79.7 | 79.8 | 77.0 | 77.8 | 44.9 | 65.0 | 61.8 | 74.9 | 67.1 | **70.5** |
| iOD (75) (Kj et al., 2021) | 40.7 | 40.9 | 28.7 | 19.1 | 23.8 | 61.6 | 56.1 | 38.8 | 23.6 | 47.5 | 18.7 | 40.1 | 40.2 | 41.5 | 39.8 | 9.1 | 40.6 | 32.4 | 41.9 | 47.6 | 36.6 |
| iOD + Ours (75) | 45.0 | 43.9 | 32.1 | 23.7 | 28.0 | 56.4 | 58.3 | 40.2 | 25.6 | 45.6 | 28.7 | 37.2 | 47.2 | 42.4 | 40.6 | 15.3 | 43.1 | 29.2 | 45.7 | 48.2 | **38.8** |

| 19 + 1 setting | aero | cycle | bird | boat | bottle | bus | car | cat | chair | cow | table | dog | horse | bike | person | plant | sheep | sofa | train | tv | mAP |
|---|---|---|---|---|---|---|---|---|---|---|---|---|---|---|---|---|---|---|---|---|---|
| OW-DETR (50) (Gupta et al., 2022) | 70.5 | 77.2 | 73.8 | 54.0 | 55.6 | 79.0 | 80.8 | 80.6 | 43.2 | 80.4 | 53.5 | 77.5 | 89.5 | 82.0 | 74.7 | 43.3 | 71.9 | 66.6 | 79.4 | 62.0 | 70.2 |
| ROSETTA (50) (Yang et al., 2022a) | 75.3 | 77.9 | 65.3 | 56.2 | 55.3 | 79.6 | 84.6 | 72.9 | 49.2 | 73.7 | 68.3 | 71.0 | 78.9 | 77.7 | 80.7 | 44.0 | 69.6 | 68.5 | 76.1 | 68.3 | 69.6 |
| iOD (50) (Kj et al., 2021) | 78.2 | 77.5 | 69.4 | 55.0 | 56.0 | 78.4 | 84.2 | 79.2 | 46.6 | 79.0 | 63.2 | 78.5 | 82.7 | 79.1 | 79.9 | 44.1 | 73.2 | 66.3 | 76.4 | 57.6 | 70.2 |
| iOD + Ours (50) | 78.3 | 77.9 | 73.3 | 57.4 | 59.0 | 80.1 | 84.7 | 80.9 | 50.0 | 81.0 | 64.9 | 82.0 | 82.9 | 80.1 | 77.7 | 46.9 | 72.8 | 64.5 | 74.8 | 61.4 | **71.8** |
| iOD (75) (Kj et al., 2021) | 35.9 | 44.7 | 31.6 | 22.4 | 26.9 | 52.0 | 56.5 | 38.7 | 21.6 | 48.4 | 21.2 | 35.9 | 37.9 | 30.7 | 38.7 | 17.2 | 38.5 | 34.2 | 40.7 | 46.6 | 36.0 |
| iOD + Ours (75) | 40.9 | 45.0 | 38.7 | 23.2 | 32.0 | 56.2 | 62.6 | 40.3 | 24.9 | 48.9 | 28.7 | 46.3 | 41.8 | 42.4 | 41.5 | 18.4 | 44.0 | 37.9 | 44.6 | 48.2 | **40.3** |

Table 3: Performance analysis of OSOD. We report close-set performance ($mAP_{\mathcal{K}}$) on VOC, and both close-set ($mAP_{\mathcal{K}}$) and open-set (WI, AOSE, $AP_{\mathcal{U}}$) performance of different methods on VOC-COCO-{20, 40, 60}. Here, '20', '40', and '60' indicate that the testing COCO images separately contain 20, 40, and 60 non-VOC classes.

| Method | VOC | VOC-COCO-20 | | | | VOC-COCO-40 | | | | VOC-COCO-60 | | | |
|---|---|---|---|---|---|---|---|---|---|---|---|---|---|
| | $mAP_{\mathcal{K}\uparrow}$ | $WI_\downarrow$ | $AOSE_\downarrow$ | $mAP_{\mathcal{K}\uparrow}$ | $AP_{\mathcal{U}\uparrow}$ | $WI_\downarrow$ | $AOSE_\downarrow$ | $mAP_{\mathcal{K}\uparrow}$ | $AP_{\mathcal{U}\uparrow}$ | $WI_\downarrow$ | $AOSE_\downarrow$ | $mAP_{\mathcal{K}\uparrow}$ | $AP_{\mathcal{U}\uparrow}$ |
| FR-CNN (Ren et al., 2015) | 80.10 | 18.39 | 15118 | 58.45 | 0 | 22.74 | 23391 | 55.26 | 0 | 18.49 | 25472 | 55.83 | 0 |
| FR-CNN[†] (Ren et al., 2015) | 80.01 | 18.83 | 11941 | 57.91 | 0 | 23.24 | 18257 | 54.77 | 0 | 18.72 | 19566 | 55.34 | 0 |
| PROSER (Zhou et al., 2021) | 79.68 | 19.16 | 13035 | 57.66 | 10.92 | 24.15 | 19831 | 54.66 | 7.62 | 19.64 | 21322 | 55.20 | 3.25 |
| ORE (Joseph et al., 2021) | 79.80 | 18.18 | 12811 | 58.25 | 2.60 | 22.40 | 19752 | 55.30 | 1.70 | 18.35 | 21415 | 55.47 | 0.53 |
| DS (Miller et al., 2018) | 80.04 | 16.98 | 12868 | 58.35 | 5.13 | 20.86 | 19775 | 55.31 | 3.39 | 17.22 | 21921 | 55.77 | 1.25 |
| OpenDet (Han et al., 2022) | 80.02 | 14.95 | 11286 | 58.75 | 14.93 | 18.23 | 16800 | 55.83 | 10.58 | 14.24 | 18250 | 56.37 | 4.36 |
| OpenDet + Ours | **80.18** | **12.23** | **10160** | **59.88** | **15.89** | **14.38** | **13580** | **57.51** | **11.82** | **12.08** | **16681** | **57.25** | **5.07** |

indicates that existing detection methods are easily affected by OOD objects. Meanwhile, based on FPR95 and AUROC metrics, our method significantly outperforms the compared methods. This not only demonstrates that synthesizing virtual features is feasible for OOD-OD but also shows that our MPD method could perform content-oriented feature synthesis effectively. Fig. 3 shows some detection results. Compared with the baseline method (Du et al., 2022c), our method accurately localizes and recognizes OOD objects, which further demonstrates the superiorities of our method.

## 4.2 PERFORMANCE ANALYSIS OF IOD AND OSOD

To further demonstrate the effectiveness of our method, we verify our method on two different tasks, i.e., IOD (Kj et al., 2021) and OSOD (Han et al., 2022). We directly plug our method into the two state-of-the-art methods (Kj et al., 2021; Han et al., 2022). Meanwhile, we do not utilize the uncertainty loss. The training and testing processes are the same as the two baselines (Kj et al., 2021; Han et al., 2022). Table 2 and 3 show the detection results. We can see that plugging our method into the two baseline methods improves their performance significantly. This further demonstrates that leveraging the phase information indeed could synthesize expected features, which strengthens the discrimination ability of the object detector.

## 4.3 ABLATION AND VISUALIZATION ANALYSIS

In this section, we utilize PASCAL VOC as the ID data for training and MS-COCO as the OOD data to perform an ablation analysis of our method.

**Analysis of MPD.** Our method mainly includes the module for synthesizing OOD features, the module for feature enhancement, and that for generating augmented features. In Table 4, we make an ablation experiment of our method. We can see that only synthesizing virtual OOD features could improve the ability of detecting OOD objects. This shows that leveraging the phase is beneficial

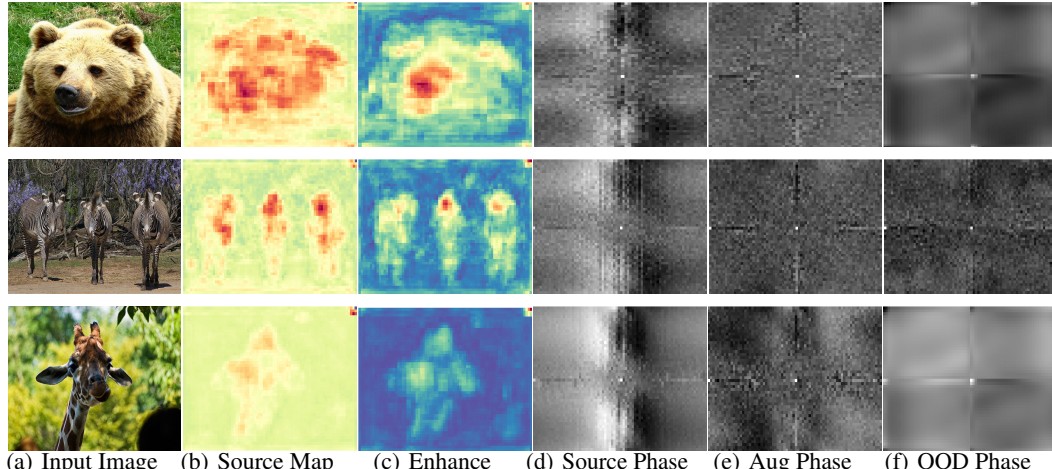

| (a) Input Image | (b) Source Map | (c) Enhance | (d) Source Phase | (e) Aug Phase | (f) OOD Phase |

Figure 4: Visualization of the Source map $F_0$, Enhanced map $E$ (i.e., $E = \Psi([F_0, F_0 - F_T])$), Source phase $P_0$, Augmented phase $\tilde{P}_0$, and OOD phase $\hat{P}_0$ based on the OOD data (MS-COCO).

for obtaining the features owning different content. Besides, we observe that performing feature augmentation and enhancement further boosts the performance, which demonstrates that our method could effectively synthesize the features enhancing the discrimination.

**Reverse iteration number $T$.** During reverse processes, we continually repeat the same operations to obtain virtual OOD features and augmented features. Here, we analyze the impact of the iteration number. We do not change our method and training details. Table 5 shows the results. Through multiple average operations, the content in the phase is significantly weakened. It is difficult to utilize a small number of iterations to acquire proper features involving plentiful content. We can see that using more iterations is beneficial for synthesizing expected OOD and augmented features, strengthening the discrimination ability of the object detector.

Table 4: Ablation analysis of MPD for unsupervised OOD-OD. 'OOD' and 'Aug' separately indicate synthesizing virtual ood features and augmented features. 'Enhance' represents $E = \Psi([F_0, F_0 - F_T])$.

| OOD | Aug | Enhance | FPR95↓ | AUROC↑ | mAP↑ |
|---|---|---|---|---|---|
| ✓ | | | 44.15% | 89.44% | 48.9% |
| ✓ | ✓ | | 42.26% | 89.91% | 49.1% |
| ✓ | | ✓ | 43.58% | 89.62% | **49.2%** |
| ✓ | ✓ | ✓ | **41.28%** | **90.54%** | **49.2%** |

**Analysis of modulated operations.** In equation 4, to synthesize augmented features effectively, we exploit the modulated operation to perform a transformation for the output of the OOD branch. Here, we use a newly-designed U-Net model to replace the modulated operation. We observe that the performance decreases severely, e.g., the FPR95 value is increased by around 5.9%. The reason may be that using a new U-Net module increases the parameters, leading to overfitting. More ablation experiments are shown in Appendix.

Table 5: Analysis of the iteration number $T$ in the reverse stage.

| $T$ | FPR95↓ | AUROC↑ | mAP↑ |
|---|---|---|---|
| 1 | 45.38% | 89.31% | 48.9% |
| 2 | 43.94% | 89.52% | 49.1% |
| 4 | **41.28%** | **90.54%** | **49.2%** |

**Visualization analysis.** In Fig. 4, we show some visualization examples. Compared with the original feature map $F_0$, the enhanced feature map $E$ contains stronger object-related information. This shows that the object-related content in the averaged phase is weakened during the forward process. Besides, the phase of OOD features is significantly different from that of original features and augmented features, demonstrating that our MPD method could effectively synthesize expected features that differ from the content of the original features and alleviate the impact of lacking OOD data.

## 5 CONCLUSION

For unsupervised OOD-OD, we focus on leveraging the phase information to perform content-oriented feature synthesis and propose a new method, i.e., modulated phase diffusion. Specifically, the forward process gradually performs Gaussian Average on the phase. Then, two different reverse processes are separately designed to synthesize expected virtual OOD features and augmented features. Extensive experimental results on three different tasks demonstrate the effectiveness of our method.

## ACKNOWLEDGEMENT

This work is supported in part by the National Key R&D Program of China (No. 2023YFC3305600), Joint Fund of Ministry of Education of China (8091B022149, 8091B02072404), National Natural Science Foundation of China (62132016, 62171343, 62071361, and 62102293), and Fundamental Research Funds for the Central Universities (ZDRC2102).

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

# A  APPENDIX

For unsupervised OOD-OD, to overcome the limitation of lacking OOD data for supervision and improve the discrimination ability of the object detector, this paper proposes a new feature-level generator, i.e., Modulated Phase Diffusion, aiming to exploit the phase information to perform content-oriented feature synthesis. In the appendix, we provide implementation details, additional analyses, various ablation studies, and more visualization results.

## A.1  EXPERIMENTAL SETUP

**Implementation Details.** We utilize Faster R-CNN (Ren et al., 2015) with RoI-Alignment layer (He et al., 2017) as the basic detection model. ResNet-50 (He et al., 2016) is taken as the backbone. For the forward process, we perform four Gaussian Average operations. $\sigma$ in equation 1 is set to 0.8. For the reverse stage, the encoder and decoder in the U-Net model $\epsilon_\theta$ all consist of three convolutional layers. All the experiments are trained using the standard SGD optimizer with a learning rate of 0.02.

**Datasets.** For unsupervised OOD-OD, we adopt PASCAL VOC (Everingham et al., 2010) and Berkeley DeepDrive (BDD-100k) (Yu et al., 2020) as the ID data for training. Meanwhile, MS-COCO (Lin et al., 2014) and OpenImages (Kuznetsova et al., 2020) are taken as the OOD datasets to evaluate the trained model. And the OOD datasets are manually examined to guarantee they do not contain ID categories. Besides, PASCAL-VOC includes the following categories: Person, Car, Bicycle, Boat, Bus, Motorbike, Train, Airplane, Chair, Bottle, Dining Table, Potted Plant, TV, Sofa, Bird, Cat, Cow, Dog, Horse, Sheep. BDD-100k contains the following classes: Pedestrian, Rider, Car, Truck, Bus, Train, Motorcycle, Bicycle, Traffic light, Traffic sign.

For IOD, we follow the standard evaluation protocol (Kj et al., 2021) and evaluate our method on PASCAL VOC (Everingham et al., 2010). We initially learn 10, 15, or 19 base classes, and then introduce 10, 5, or 1 new classes as the second task. Finally, for OSOD, we follow the work (Han et al., 2022) and utilize 20 VOC classes and 60 non-VOC classes in COCO to evaluate our method under different open-set conditions.

**Metrics.** For OOD-OD, we report: (1) the false positive rate (FPR95) of OOD objects when the true positive rate of ID objects is at 95%; (2) the area under the receiver operating characteristic curve (AUROC); (3) mean average precision (mAP). For OSOD, we use Wilderness Impact (WI) (Dhamija et al., 2020) to measure the degree of unknown objects misclassified to known classes. And we also use Absolute Open-Set Error (AOSE) (Miller et al., 2018) to count the number of misclassified unknown objects.

## A.2  MORE EXPERIMENTAL DETAILS OF IOD AND OSOD

To further demonstrate the effectiveness of our method, we verify our method on IOD and OSOD. Here, we directly plug our method into two baseline methods and do not calculate the uncertainty loss. The training details are the same as the baselines.

To effectively exploit the synthesized virtual OOD features, we train a binarized classifier, i.e., the output of the known category is 1, and the output of the virtual OOD features is 0. Meanwhile, we still utilize a $KL$-divergence to constrain the prediction consistency between the original features and augmented ones. By these operations and minimizing the cross-entropy loss, the discrimination ability of the object classifier could be further strengthened.

## A.3  FURTHER DISCUSSION OF VISUALIZATION RESULTS

To alleviate the impact of lacking OOD data, we focus on phase-based content-oriented feature synthesis. Fig. 5 shows more visualization examples. We can see that compared with the original feature map $F_0$, the enhanced feature map $E$ contains stronger object-related information and weaker background-relevant information. This shows that after the multiple averaging operations, the object-related content of the forward output is gradually damaged, meeting the purpose of the forward diffusion. Besides, we can observe that the OOD phase information is significantly different from the phase of the augmented features. This further indicates that the content of the synthesized virtual

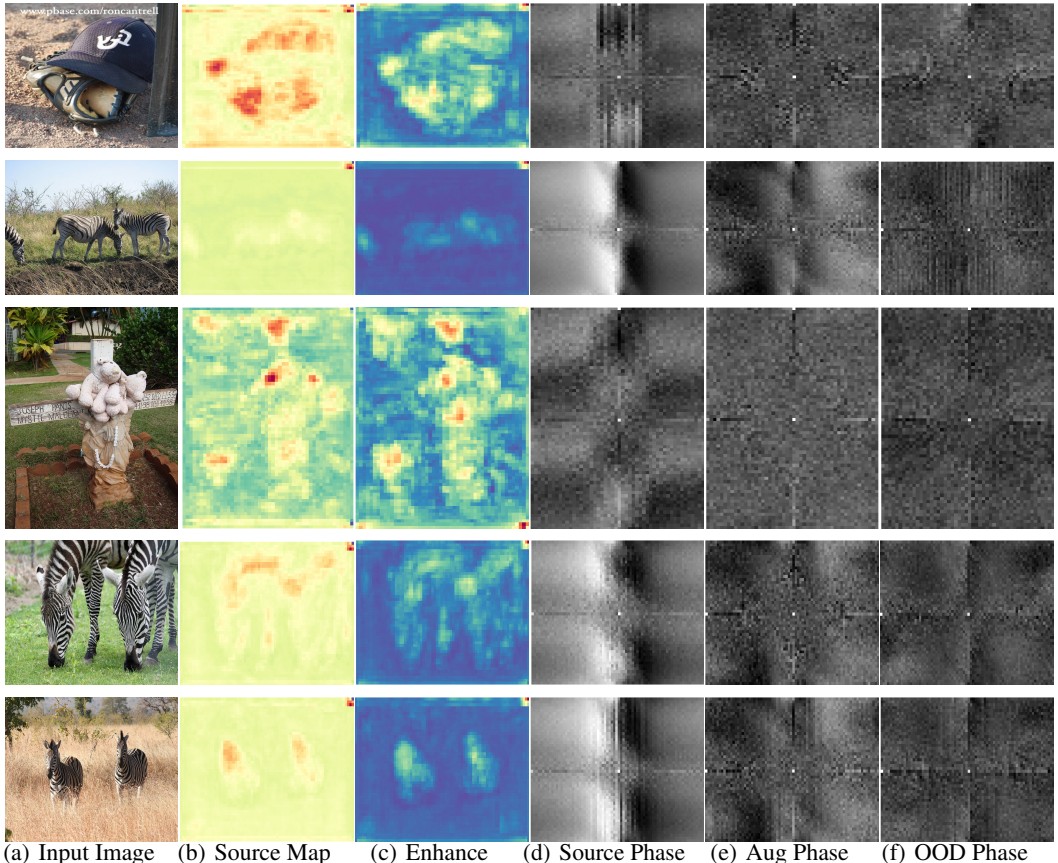

(a) Input Image    (b) Source Map    (c) Enhance    (d) Source Phase    (e) Aug Phase    (f) OOD Phase

Figure 5: Visualization of the Source map $F_0$, Enhanced map $E$ (i.e., $E = \Psi([F_0, F_0 - F_T])$), Source phase $P_0$, Augmented phase $\tilde{P}_0$, and OOD phase $\hat{P}_0$ based on the OOD data (MS-COCO).

OOD features differs from that of the input features, which attains the motivation of this paper and demonstrates the effectiveness of our method.

### A.4   MORE ABLATION EXPERIMENTS OF HYPER-PARAMETERS AND MPD METHOD

For our method, we utilize the hyper-parameter $\lambda$ for the $KL$-divergence loss (equation 7), the hyper-parameter $\alpha$ for the loss $\mathcal{L}_{MPD}$ (equation 9), and the hyper-parameter $\tau$ for the loss $\mathcal{L}_{\text{uncerty}}$ (equation 9). Since the uncertainty loss $\mathcal{L}_{\text{uncerty}}$ is directly related to the current task, the value of $\tau$ should be set larger than $\lambda$ and $\alpha$. Meanwhile, if $\lambda$ and $\alpha$ are set to a small value, the role of the two corresponding losses will be weakened in optimization. Thus, it is meaningful to set proper values for these hyper-parameters. Here, we take PASCAL VOC as the ID data and MS-COCO as the OOD data to perform an ablation analysis of hyper-parameters. And we only change these hyper-parameters and keep other modules unchanged.

**Analysis of $\lambda$.** The hyper-parameter $\lambda$ in equation 7 is to balance the detection loss and the loss that aims to minimize the $KL$-divergence between the prediction probabilities from $O_{\text{in}}$ and $O_{\text{aug}}$. In the experiments, we observe that when $\lambda$ is set to 0.01, 0.001, and 0.0001, the performance of FPR95 is 42.94%, 41.28%, and 42.16%.

**Analysis of $\alpha$.** The goal of the hyper-parameter $\alpha$ in equation 9 is to weigh the importance of the module of MPD. In the experiments, we find that when $\alpha$ is set to 0.01, 0.001, and 0.0001, the corresponding FPR95 performance is 43.25%, 41.28%, and 41.92%.

**Analysis of $\tau$.** In this paper, the hyper-parameter $\tau$ in equation 9 is to constrain the uncertainty loss $\mathcal{L}_{\text{uncerty}}$. In the experiments, we observe that when $\tau$ is set to 0.5, 0.1, and 0.01, the corresponding performance of FPR95 is 43.89%, 41.28%, and 43.13%.

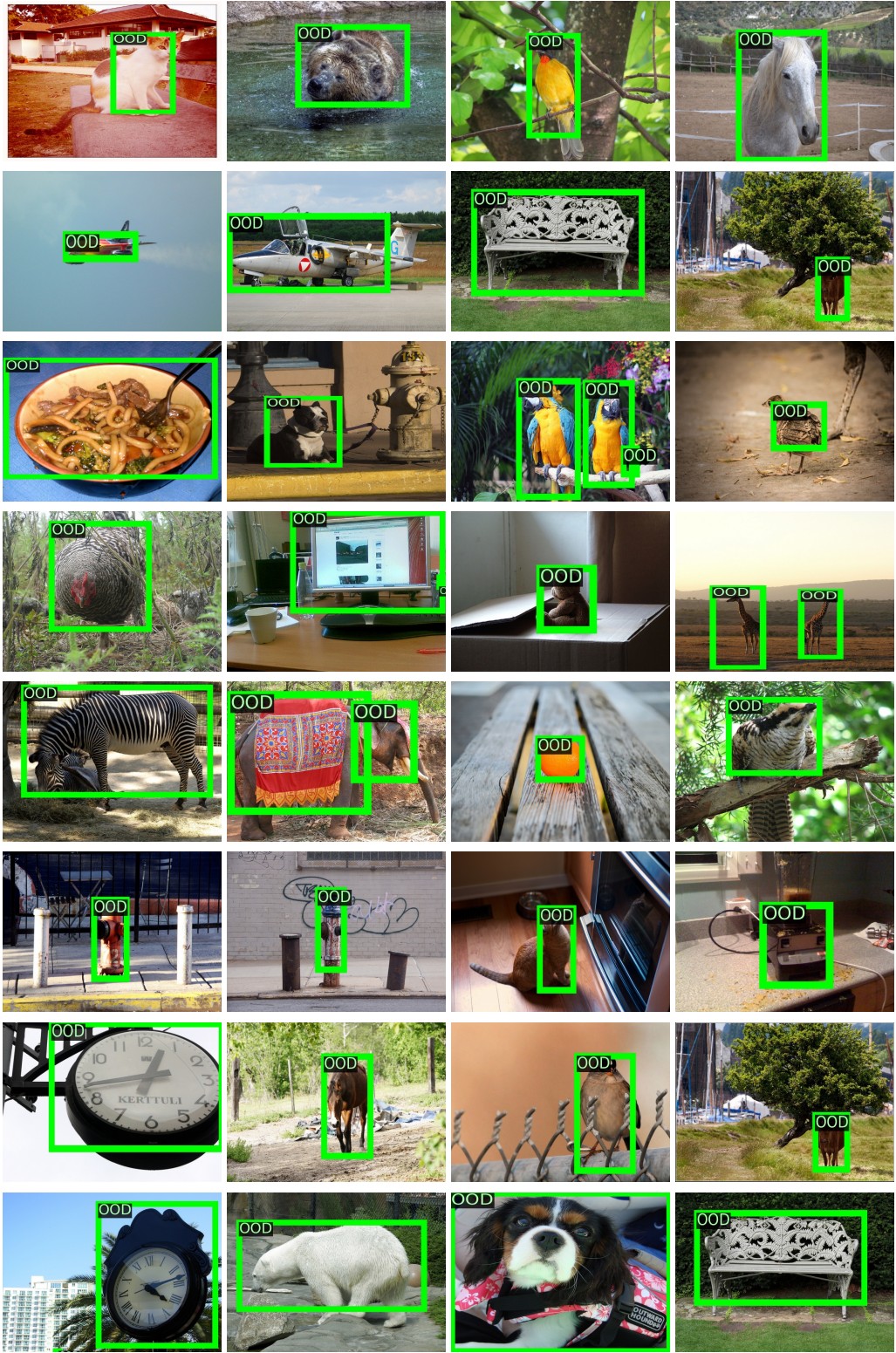

Figure 6: OOD detection examples based on our method. Here, we use BDD-100k as the in-distribution data and MS-COCO as the OOD data. We can see that our method accurately distinguishes OOD objects, which shows the effectiveness of our method.

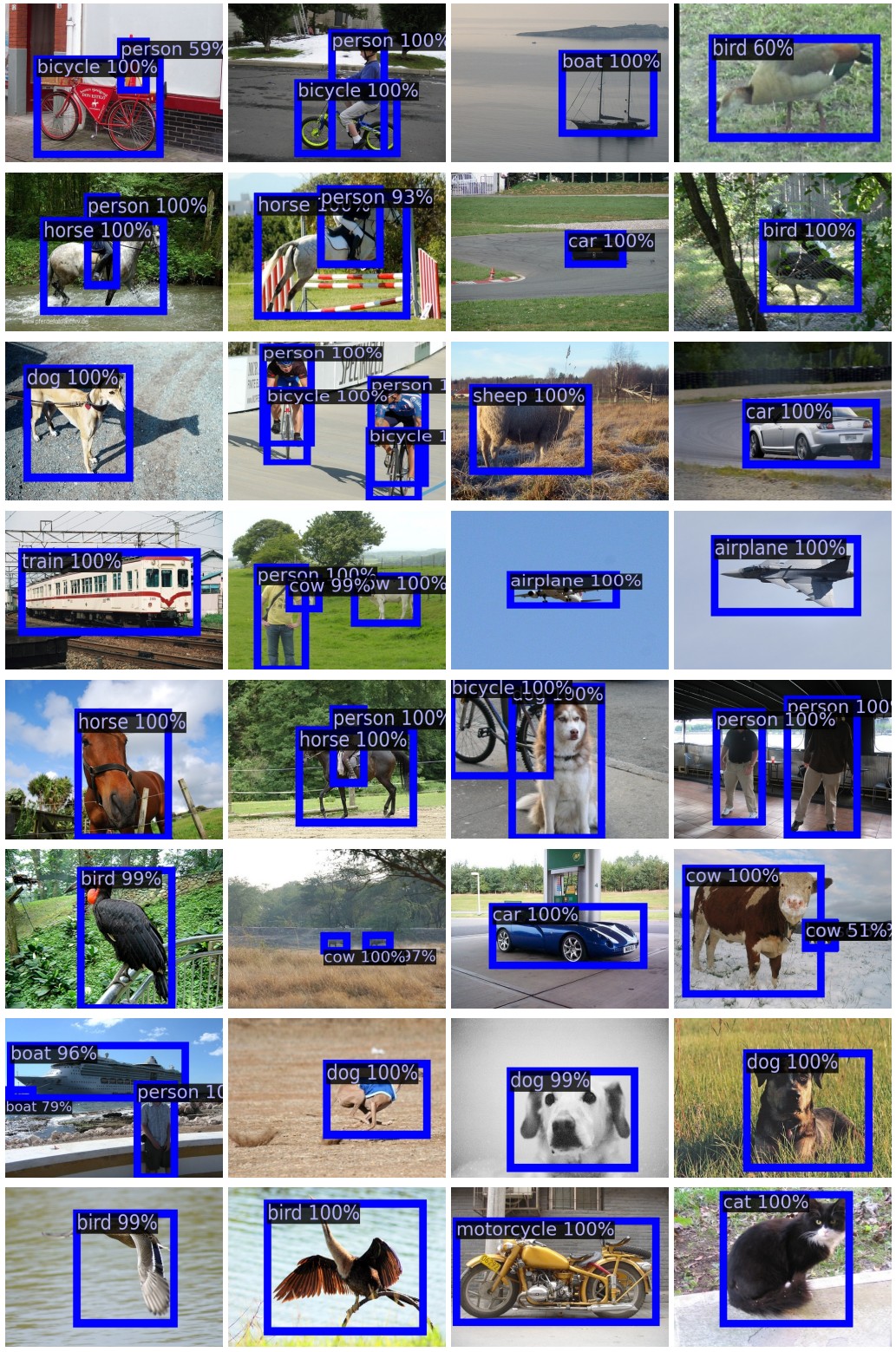

Figure 7: Detection results based on PASCAL VOC. We can see that our method accurately localizes and recognizes objects in these images, e.g., the dog, car, cow, and person, which shows that our method is effective for in-distribution data.

Table 6: Definitions of notations used in our method.

| Notations | Definition |
|---|---|
| $F_0$ | The feature map extracted by the backbone network. |
| $A, P_0$ | The corresponding amplitude and phase of $F_0$. |
| $\mathcal{G}(\sigma)$ | Gaussian Kernel with the variance $\sigma$. |
| $P_t$ | The convolutional output of the $t$-th Gaussian Average operation. |
| $\epsilon_\theta(\cdot, \cdot)$ | The learned U-Net model. |
| $\hat{D}_{t-1}$ | The predicted result of the U-Net model. |
| $\hat{P}_{t-1}$ | The decomposed phase based on $\hat{D}_{t-1}$. |
| $\hat{F}_{t-1}$ | The IFFT output of $\hat{P}_{t-1}$ and $A$. |
| $\hat{F}_0$ | The synthesized virtual OOD map. |
| $\mathcal{D}_{t-1}$ | The output of the modulation mechanism. |
| $\mathcal{P}_{t-1}$ | The decomposed phase based on $\mathcal{D}_{t-1}$. |
| $\tilde{F}_0$ | The augmented feature map. |

**Analysis of the Kernel Size.** In equation 1, during the forward process, we exploit the Gaussian Kernel to perform the average operation. Here, we make an ablation analysis of the kernel size. And other modules are kept unchanged. Taking PASCAL VOC as ID data for training and MS-COCO as OOD data for evaluation, when the kernel size is separately set to $3 \times 3$, $5 \times 5$, and $7 \times 7$, the corresponding performance is 41.76%, 41.28%, and 42.84%.

In Fig. 6 and 7, we show more detection results from our method. We can see that our method could distinguish ID objects and OOD objects accurately, which further demonstrates the effectiveness of our phase-based diffusion method.

## A.5 DEFINITIONS OF NOTATIONS

Table 6 gives the definitions of notations used in our method.

## A.6 FURTHER DISCUSSION OF OUR METHOD

For unsupervised OOD-OD, since there is no OOD data available, one feasible solution is to leverage the known ID data to synthesize expected OOD features that differ from the content of ID features. To this end, in this paper, we explore leveraging the phase component to generate expected OOD features, which is instrumental in improving the ability of discriminating OOD from ID objects. In the following, we will give more discussions about our method.

**On performing Gaussian Average.** During the forward process, existing diffusion methods (Ho et al., 2020; Luo, 2022) are fixed to a Markov chain that gradually adds Gaussian noise to the input. And a notable property of the forward process is that it admits sampling the diffusion results at an arbitrary timestep $t$ due to the characteristic of Gaussian noise (Ho et al., 2020).

Since our method is to leverage the phase components to generate expected features, directly adding noise to the phase may rapidly destroy the content in the phase, affecting the performance of the diffusion model. To this end, we replace adding noise with gradually performing Gaussian Average. Compared with globally adding noise, Gaussian Average is a local operation and could reduce the damage to the content. The overall forward process can be described as follows:

$$q(P_t|P_{t-1}) = P_{t-1} * \mathcal{G}(\sigma), \tag{11}$$

where $P_t$ represents the $t$-th output phase of the forward diffusion process.

Meanwhile, the forward process owns a notable property that it admits averaging the phase $P_t$ at an arbitrary timestep $t$:

$$P_t = P_{t-1} * \mathcal{G}(\sigma) = (P_{t-2} * \mathcal{G}(\sigma)) * \mathcal{G}(\sigma) = P_0 * \mathcal{G}(\sigma) * \cdots * \mathcal{G}(\sigma) = P_0 * \mathcal{G}(\sigma_s), \tag{12}$$

where $\sigma_s = \sum_{k=1}^{t} \sigma$. Like traditional diffusion methods (Ho et al., 2020; Luo, 2022), this property admits acquiring the diffusion results at an arbitrary timestep $t$. Experimental results on multiple datasets demonstrate the effectiveness of this operation.

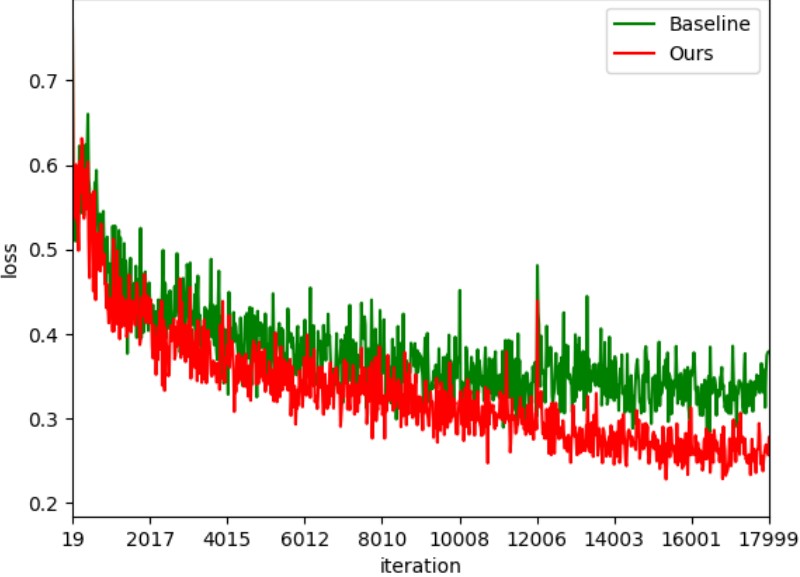
Figure 8: The training loss of the baseline (Du et al., 2022c) and our method.

**On the reverse process.** The reverse process aims to generate virtual OOD feature $\hat{F}_0$ from an averaged result $F_T$:

$$p_\theta(\hat{F}_{T:0}) := p(\hat{F}_T) \prod_{t=T}^{1} p_\theta(\hat{F}_{t-1}|\hat{F}_t), \tag{13}$$

where $\hat{F}_T = F_T$. An U-Net network is taken as the decoder to predict the output $\hat{F}_t$ involving OOD-related information.

The reverse loss $\mathcal{L}_{\text{reverse}}$ can be represented as follows:

$$\mathcal{L}_{\text{reverse}} = \mathbb{E}_q[D_{KL}(q(F_T|F_0)||p(\hat{F}_T)) - \sum_{t>1} D_{KL}(q(F_{t-1}|F_t)||p_\theta(\hat{F}_{t-1}|\hat{F}_t)) - \log p_\theta(\hat{F}_0|\hat{F}_1)], \tag{14}$$

where the first term is a constant during training and can be ignored. By minimizing the last two terms, the gap between the synthesized OOD features and ID features could be enlarged, promoting the synthesized OOD features to contain plentiful content that differs from the ID input. Concretely, for the second loss term, we explore making the output of the decoder gradually contain less ID-relevant content to promote the decoder to own the ability of generating OOD samples (as shown in equation 3). Meanwhile, for the last loss term, we maximize the loss $\mathcal{L}_{\text{dis}}$ to further enlarge the gap between the OOD and ID features. Extensive experimental results and visualization analysis demonstrate that our method could synthesize expected features effectively, improving the ability of discriminating OOD objects.

## A.7 VISUALIZATION OF THE TRAINING LOSS CURVE

In Fig. 8, we show the training loss curves of the baseline (Du et al., 2022c) and our method. The decreasing speed of the loss value is around consistent with the baseline method. We can see that the loss curve of our method is significantly lower than that of the baseline, which further demonstrates the effectiveness of our method.

