# OpenReview forum: "Modulated Phase Diffusor: Content-Oriented Feature Synthesis for Detecting Unknown Objects"
_ICLR.cc/2024/Conference — ICLR 2024 poster_

### Official Review · Reviewer_A444 · 2023-10-29

**Soundness:** 3 good
**Presentation:** 3 good
**Contribution:** 3 good
**Rating:** 6
**Confidence:** 3

**Summary:**

This manuscript attempts to use the Diffusion model to solve the unsupervised out-of-distribution object detection (OOD-OD) task. This method uses two inverse processes to synthesize the phase information of unknown and known samples, respectively. The experimental results validate the effectiveness of the method.

**Strengths:**

1. Gradual phase averaging: Instead of adding noise, MPD gradually performs Gaussian averaging on the phase of extracted features. This helps to prevent rapid loss of content in the phase, ensuring that important information is preserved during the process.

2. Experimental superiority: MPD has demonstrated superior performance in various tasks, including OOD-OD, incremental object detection, and open-set object detection. The experimental results validate the effectiveness and advantages of the MPD method in promoting safe deployment of object detectors.

**Weaknesses:**

The idea of replacing Gaussian noise in the diffusion process with Gaussian average operation seems to be an experimental result, lacking theoretical explanation and formula derivation.

**Questions:**

1. Why choose to generate phase instead of the original image or amplitude? What are the phase advantages?
2. Why choose a 5x5 kernel for the Gaussian average? Has the author tried other types or sizes of kernels?
3. If Gaussian noise is replaced by the Gaussian average, is the diffusion model still valid? Can you provide the formula derivation?
4. The description of the OOD phase in Figure 2 is confusing, the meaning of approximately equal and not equal symbols is unclear.
5. What are the shortcomings of this method? Does its training and reasoning time have any advantages compared to previous methods?

---

> ### Author Response · Authors · 2023-11-17
> **Thanks for your helpful comments.**
>
> Q1: Theoretical Analysis of Gaussian Average Operation
>
> A: Thanks for your helpful comment. Here, we give some theoretical analysis of Gaussian Average Operation.
>
> We first introduce the background on diffusion models. For each training data $\mathbf x_0 \sim q_0(\mathbf x)$, a forward process of the diffusion models is defined from the following Markov chain:
>
> $\mathbf x_i = \sqrt{1-\beta_i}\mathbf x_{i-1} + \sqrt{\beta_i}\mathbf z_{i}, \ \ i=1,...,N$, (1)
>
> where $z_i\sim \mathcal N(0,\mathbf I)$, and $\{\beta_i\}_{i=1}^N$ is a pre-defined noise schedule. Next, we generalize this process in a rorated coordinate system. We first define an orthogonal matrix ${\rm U}$, and subsequently some vector rotated by the matrix as $\bar{\rm x} := {\rm U}^T {\rm x}$.
>
> Then, a generalized forward diffusion process is defined as the following Markon chain:
>
> $q(\bar{\rm x}{\tiny{i}} |\bar{\rm x}{\tiny{i-1}}) = \mathcal{N}(\bar{\rm x}{\tiny{i}}; ({\rm I} - {\rm B}{\tiny{i}})^{\frac{1}{2}}\bar{\rm x}{\tiny{i-1}}, {\rm B}{\tiny{i}}{\rm I})$, (2)
>
> where ${\rm B}_i$ is a diagonal matrix that defines the noise schedule of the process. It is worth noting that Eq. (2) is a generalized version of the standard diffusion, as standard diffusion is retrieved when we set ${\rm U} = {\rm I}$, and ${\rm B}_i = \beta_i {\rm I}$.
>
> Next, we analyze the proposed Gaussian Average Diffusion:
>
> While the choice of the rotation matrix ${\rm U}$ and the noise schedule ${\rm B}_i$ are flexible, we propose to replace adding noise with effective Gaussian Average. Since Gaussian Average is a linear operation, it can be approximated as a matrix multiplication using a circular symmetric matrix $\mathbf W$. At this time, the process of Gaussian Average is defined as follows:
>
> $q({\rm x}{\tiny{i}}|{\rm x}{\tiny{i-1}}) = \mathcal{N}({\rm x}{\tiny{i}}; \sqrt{1-\beta{\tiny{i}}}{\rm W}{\tiny{i}}{\rm x}{\tiny{i-1}}, {\rm C}{\tiny{i}})$,   (3)
>
> where we set ${\rm C}_i = {\rm I} - (1-\beta_i){\rm W}_i^2$ to ensure the process preserves unit variance. Meanwhile, Eq. (3) can also be written as follows:
>
> $\mathbf x_i = \mathbf x_{i-1} -\mathbf H(\mathbf x_{i-1}, i-1) +  {\rm C}_i^{\frac{1}{2}}{\rm z}_i$, (4)
>
> where $\mathbf H(\mathbf x_{i}, i) = \mathbf x_{i} - \sqrt{1-\beta_{i+1}}{\rm W}_{i+1}{\rm x}_i$ is an unnormalized Gaussian high-pass filter. Thus, the forward process gradually destroys high frequencies. To match the definition of the generalized diffusion, we propose to factor the symmetric matrix ${\rm W}$ by eigenvalue-decomposition ${\rm W} = \tilde{\rm U}{\rm D}\tilde{\rm U}^T$ and subsequently ${\rm W}_i = \tilde{\rm U}{\rm D}^{f(i)}\tilde{\rm U}^T$, where $f(i)$ determines a blur schedule.
>
> Meanwhile, when ${\rm B}_i = {\rm I} - (1-\beta_i) {\rm D}^{2f(i)}$ and ${\rm U} = \tilde{\rm U}$, Eq. (3) is equivalent to Eq. (2).
>
> Besides, in Sec. A.6 of Appendix, we also give some analysis about our method. These analyses all demonstrate that the diffusion with Gaussian Average could be used to synthesize specific features effectively, which improves the discrimination ability.

---

> ### Author Response · Authors · 2023-11-17
> **We will modify our paper carefully according to your helpful comments.**
>
> Thanks for your recognition of our work. We will modify this paper carefully according to your valuable comments.
>
> Q2: The reason of generating phase
>
> A: For unsupervised OOD-OD, since there is no OOD data available, we could not ensure the generated images contain expected OOD contents. Meanwhile, compared with generating features, generating images may require more computational costs. And the generated images are taken as the input for training, increasing the training costs.
>
> In the third paragraph of the Introduction Section, we have interpreted the reason of generating phase. In general, the extracted features of input images could be recognized as containing style and content information. And the content of the OOD features should be different from that of the ID features. Besides, from the frequency perspective, recent researches have shown that the amplitude and phase could be separately regarded as the style and content of the input. To this end, we pay more attention to exploiting the phase information to perform content-oriented feature synthesis, which is instrumental in addressing the key challenge of unsupervised OOD-OD.
>
> Extensive experiments and visualization analysis also demonstrate that leveraging the phase components is indeed helpful for generating expected virtual features, which is instrumental in improving the discrimination ability.
>
> Q3: On the other sizes of Gaussian kernel
>
> A: Thanks for your helpful comment. In this paper, we empirically select $5 \times 5$ kernel.
>
> In the Appendix (i.e., Sec A.4), we have made an ablation analysis of the kernel size. Taking PASCAL VOC as ID data for training and MS-COCO as OOD data for evaluation, when the kernel size is separately set to $3 \times 3$, $5 \times 5$, and $7 \times 7$, the corresponding performance is 41.76%, 41.28%, and 42.84%.
>
> Q4: On the approximately equal and not equal symbols in Figure 2
>
> A: Thanks for your valuable comment. We have revised Figure 2 and removed the approximately equal and not equal symbols. And OOD Phase in Figure 2 is replaced by OOD Feat. Phase.
>
> Q5: On the shortcomings of this method
>
> A: During training, since our method introduces the diffusion process, the training time is slightly longer than the baseline method [1], which could be considered a shortcoming. However, compared with GAN-based methods [2], the training and reasoning time is better than them.
>
> [1] Xuefeng Du, Zhaoning Wang, Mu Cai, and Yixuan Li. Vos: Learning what you don’t know by virtual outlier synthesis. ICLR, 2022.
>
> [2] Kimin Lee, Honglak Lee, Kibok Lee, and Jinwoo Shin. Training confidence-calibrated classifiers for detecting out-of-distribution samples. ICLR, 2018.

---

### Official Review · Reviewer_3rox · 2023-10-30

**Soundness:** 2 fair
**Presentation:** 1 poor
**Contribution:** 2 fair
**Rating:** 3
**Confidence:** 5

**Summary:**

This paper focuses on the OOD object detection problem and propose to detect unknown object without relying on auxiliary OOD data. This paper exploits ID data to generate OOD data by considering the phase information in frequency spectrum. A modulated phase diffusion (MPD) is designed, with some detailed forward and reverse compuation. Experiments on several tasks show the effectiveness.

**Strengths:**

+The proposed technical framework sounds good, by leveraging frequency information, U-net and augmented features.
+ Experimental results are comparable to previous SOTA models.

**Weaknesses:**

- This paper writting has a large space for improvement and not easy to follow. Although the authors presented the main motivations of this paper, there are still many places unclear. The phase information of ID features is used to generate OOD features. Since phase represents more the content, the amplititude information may be more important for different styles (OOD featuers).
-The motivation on the augmented features (ID or OOD?) is not estabilished.
-What is the difference between OOD-OD and open-set OD? The authors seem list them as different. But as I see, they are the same problem.
-The forward and reverse process are not clear due to the poor writting.
-Fig.1 and Fig.2 are redundant which evens show similar objective about the proposed MPD. Also, the designed method seems complex and not easy to follow.
-I also concern about the claim "lacking unknown data". Since OOD-OD is problem setting, it is rational to suppose some categories are unknown, such as open-set OD.
-Lacking the visualization results of phase based OOD data synthesis.

**Questions:**

1. How about the conventional unknown object detection based on a simple threshold, such as entropy based.
2. Minimizing the KL between ID and OOD is strange in Eq. 7.
3. In Eq. 9, there are many losses, which makes the training not easy.

---

> ### Author Response · Authors · 2023-11-17
> **Thanks for your valuable comments.**
>
> Thanks for your helpful comments. We will modify our paper carefully according to your valuable comments.
>
> Q1: On the motivation of the augmented features
>
> A: In the Introduction Section, we have indicated that due to the lack of OOD data for training, the challenge of unsupervised OOD-OD mainly lies in how to leverage the known ID data to enhance the ability of distinguishing OOD objects while reducing the impact on the performance of detecting ID objects.
>
> To address this challenge, we leverage the phase components to synthesize expected virtual OOD features, which alleviates the impact of lacking OOD data for training. Meanwhile, we further generate augmented ID features that are beneficial for retaining the detection performance of ID objects. Extensive experimental results and visualization analysis demonstrate that our method is indeed helpful for addressing the critical challenge of unsupervised OOD-OD.
>
> Q2: The difference between OOD-OD and open-set OD
>
> A: The difference between OOD-OD and open-set OD mainly lies in their settings. Compared with OOD-OD, open-set OD requires indicating the number of unknown categories. However, their goal is consistent, i.e., detecting unknown objects. Thus, we simultaneously verify our method on OOD-OD and open-set OD. The performance improvement demonstrates the effectiveness of our method.
>
> Q3: On the visualization results of phase
>
> A: In Fig. 4 and 5, we have shown the phase of the synthesized features. The phase of OOD features is significantly different from that of original features and augmented features, demonstrating that our MPD method could effectively synthesize expected features that differ from the content of the original features and alleviate the impact of lacking OOD data.
>
> Q4: The performance of the conventional unknown object detection
>
> A: The performance of the conventional methods is very poor. In Table 1, MSP is simple baseline method. Based on PASCAL VOC dataset, we can observe that FPR95 value is 70.99% and 73.13%, which is very poor.
>
> Q5: On minimizing the KL in Eq. 7
>
> A: In Eq. 7, we have explained the role of the KL divergence loss, i.e., constraining the prediction consistency between ID and augmented features. This loss term is beneficial for ameliorating the ability of the object classifier for discriminating ID objects.
>
> Q6: On the training process
>
> A: In Eq. 9, to ensure the effectiveness of our method, we introduce more loss terms. In the experiments, we observe that using more loss terms does not increase the training difficulty. In fig8, we show the training loss curves of VOS (baseline) and our method. The decreasing speed of the loss value is around consistent with the baseline method. Meanwhile, we can see that the loss curve of our method is significantly lower than that of the baseline, which further demonstrates the effectiveness of our method.

---

> > ### Comment · Reviewer_3rox · 2023-11-18
> > **Concerns not addressed**
> >
> > Thanks for the authors' feedback. However, many of my concerns are not addressed and mentioned.
> > For example, in Q1, the motivation is still not answered and it is not important to just describe how you do that.
> > "lacking unknow data" is also not mentioned.

---

> ### Author Response · Authors · 2023-11-18
> **Further interpretation about the concerns**
>
> Q1: Further interpretation of our motivation
>
> A: Sorry for you. Here, we give more interpretations about our motivation.
>
> In the first paragraph of Introduction Section, we have indicated that the goal of unsupervised out-of-distribution object detection (OOD-OD) is to detect unknown OOD objects during training without exploiting any auxiliary OOD data.
>
> In the second paragraph, we further indicated that due to lacking OOD data for training, the challenge of unsupervised OOD-OD mainly lies in how to only leverage the known in-distribution (ID) data to enhance the ability of distinguishing OOD objects while reducing the impact on the performance of detecting ID objects. Meanwhile, we also indicate that one feasible solution is to synthesize a series of proper virtual OOD features for supervision based on the ID data, which is conducive to promoting the
> object detector to learn a clear boundary between ID and OOD objects.
>
> To address the challenge of unsupervised OOD-OD, in this paper, we explore synthesizing expected features for supervision. Besides, OOD features could be assumed to possess the content that differs from ID features. And recent researches have shown that the phase could be regarded as the content of the input. Therefore, we pay more attention to exploiting the phase information to perform content-oriented feature synthesis.
>
> Q2: The phase information of ID features is used to generate OOD features
>
> A: Sorry for confusing you. Since the phase information has been shown to be related to the content, to synthesize expected OOD features owning different content from that of ID features, we explore generating OOD features by enlarging the phase gap between the ID and generated OOD features. Experimental results demonstrate that the phase information of ID features could be effectively used to generate proper OOD features.

---

### Official Review · Reviewer_PJVM · 2023-11-01

**Soundness:** 2 fair
**Presentation:** 3 good
**Contribution:** 2 fair
**Rating:** 6
**Confidence:** 3

**Summary:**

This paper focuses on the out-of-distribution object detection (OOD-OD) task, and proposes a method named MPD to tackle it from the frequency perspective. Following the previous method VOS that alleviates the OOD-OD problem by adaptively synthesizing virtual outliers, MPD also attempts to synthesize suitable virtual OOD features as well as generate augmented features for supervised training. Different from VOS that assumes a class-conditional multivariate Gaussian distribution of the feature space, MPD in this paper tries to add noise to the phase domain in the diffusion way. Moreover, the authors find that the Gaussian Average for processing each step is better than directly adding noise. Many experiments and ablation studies have verified that MPD is superior than previous methods for dealing with OOD-OD.

**Strengths:**

I think this paper has at least the following several major contributions:

1. The authors approach the OOD-OD problem from the phase domain of the extracted image features, which is an aspect that is interesting and rarely studied.

2. Introducing diffusion to generate different features is a new attempt in OOD. After discovering that simply and directly adding noise according to the original method was problematic, the authors proposed their own effective improvement strategies.

3. Quantitative experimental results prove the advancement of MPD in many OOD-OD benchmarks.

**Weaknesses:**

Similarly, we summarize the weaknesses of this paper as follows:

1. Actually, studying the processing of features from a phase perspective is not the first of its kind in this paper. In other words, the method [1] has proven that phase-related features are content-oriented in the DG field, which is very similar to OOD. This paper directly uses such a conclusion of DG in OOD and cannot be regarded as a complete innovation.

2. Generally speaking, there are quite a few steps in the continuous transformation of the Diffusion model, such as dozens or even hundreds of steps. The method in this paper seems to only use up to 4 interations (T=4 in Table 5). Why not try more steps? Is it because more parameters are introduced (such as the U-Net model for predicting feature maps, and two branches for generating new features in OOD) that it is inconvenient to increase T to a too large number? If so, the authors need to explain clearly how the new method MPD increases the number of parameters compared to the original basic detector, such as the used Faster R-CNN.

3. As we all know, Faster R-CNN is a classic but outdated detector. It gives a weak baseline of detection comparing to recent new ones. The actual value of OOD-OD is to achieve robust and generalizable object detection in real applications. Thus, using advanced basic detectors such as YOLOv5 [2], YOLOv8 [3], TOOD [4] and DETRs is more meaningful. And it will be important to see if the proposed MPD works or not on these superior detectors. It may not be practical to do more experiments. The authors could give similar explanations and discussions. For example, is MPD universal to these superior detectors?

[1] Decompose, Adjust, Compose: Effective Normalization by Playing With Frequency for Domain Generalization, CVPR 2023

[2] https://github.com/ultralytics/yolov5, YOLOv5 2020

[3] https://github.com/ultralytics/ultralytics, YOLOv8 2023

[4] TOOD: Task-aligned One-stage Object Detection, ICCV 2021

**Questions:**

Overall, the method proposed in this paper is innovative and effective. Please go back to the two questions I mentioned in the weaknesses of items 2 and 3. Let me shorten these two questions as below:
1. How the proposed MPD increases the number of network parameters?
2. How about using advanced basic detectors instead of the outdated Faster R-CNN?

---

> ### Author Response · Authors · 2023-11-17
> **Thanks for your helpful comments.**
>
> Thanks for your recognition of this work. We will modify this paper carefully according to your valuable comments.
>
> Q1: On the novelty of our method
>
> A: We agree with you that our method is inspired by recent observations, i.e., the phase components usually involve content-related information. In general, the goal of DG task is to transfer the model to new domains with different styles. To this end, most DG methods attempt to keep the phase components unchanged and leverage the amplitude components involving style-related information to perform style-level augmentation, which is instrumental in improving the generalization ability.
>
> However, the phase components are rarely explored. Besides, different from DG tasks, the task of unsupervised OOD is to discriminate known from unknown objects without reliance on any auxiliary data. The challenge mainly lies in how to leverage the given ID data to improve the discrimination ability. One feasible solution is to synthesize virtual OOD features. Meanwhile, we consider that the OOD features should involve the content that differs from the known ID features. To this end, we explore leverage the phase components to synthesize expected virtual OOD features to alleviate the impact of lacking OOD data.
>
> Thus, the novelty of our method mainly is that we first propose a dedicated phase-driven diffusor that leverages the phase components to separately synthesize OOD features and augmented ID features, which improves the ability of discriminating OOD objects without degrading the detection performance of ID objects.
>
> Q2： On the number of newly introduced parameters
>
> A: To generate images containing clear contents, traditional diffusion models usually require to use more iteration steps to progressively recover the clear visual contents. In this paper, our goal is to synthesize expected features. Different from generating images, the generated features only satisfy that the generated contents are conducive to improving the performance of discriminating OOD from ID objects. Thus, we do not utilize more iteration steps, which does not introduce more computational costs.
>
> Compared with the original base detector, i.e., Faster R-CNN, our method introduces around 26M parameters.
>
> Q3: On applying our method to more advanced detectors
>
> A: Thanks for your helpful advice. To address the challenge of unsupervised OOD-OD, i.e., leveraging the known ID data to enhance the ability of distinguishing OOD objects while reducing the impact on the performance of detecting ID objects, our MPD method aims to leverage the phase components to synthesize virtual OOD features and augmented ID features. Therefore, our method could be applied to both Faster R-CNN and other advanced detectors.
>
> Here, we make an experiment, i.e., plugging our method into YOLOv5. We separately take PASCAL VOC and MS-COCO as the ID dataset and OOD dataset. We observe that plugging our method reduces FPR95 performance by around 7.38%. This further shows that synthesizing virtual features for unsupervised OOD-OD is an effective solution. Meanwhile, our MPD method could be applied to other detectors and improve their performance of discriminating OOD objects.

---

> > ### Comment · Reviewer_PJVM · 2023-11-20
> > **Questions about A1 and A2**
> >
> > To A1: OOD detection belongs to the DG task. Please refer to the survey paper [4]. OOD detection is a special application case of DG. MPD in this paper directly adopts work [1] that proposed a general DG method, so I think this discounts its novelty
> >
> > [4] Domain Generalization: A Survey, TPAMI 2022
> >
> > To A2: The authors still did not explain why only 2 to 4 steps of diffusion operations are enough to meet the purpose of synthesizing ideal virtual features. There is no experimental data in the paper to support this conclusion. Or does this conclusion come from elsewhere? Then please give the source of the citation.

---

> ### Author Response · Authors · 2023-11-20
> **Thanks for your helpful comments.**
>
> Q1: Further discussion of the novelty
>
> A: Thanks for your helpful comments. In this paper, our method does not directly adopt the work [1]. Instead, our method is based on the recent observation, i.e., the phase components usually describe the content information. To address the challenge of unsupervised OOD-OD, we explore generating expected virtual OOD features. Meanwhile, to synthesize effective OOD features, we follow an assumption, i.e.,  the content in the OOD features is different from that in the ID features. Thus, we first design a dedicated phase-driven diffusor to separately synthesize virtual OOD and augmented ID features, which is the key contribution of our work.
>
> Q2: On the iteration number of the reverse process
>
> A: Thanks for your valuable comment. Here, we make an experiment about the iteration number:
>
>
> | Number $T$ | FPR95 $\downarrow$ |
> | :-----| ----: |
> | 1 | 45.38% |
> | 2 | 43.94% |
> | 4 | 41.28% |
> | 8 | 41.34% |
> | 16 | 41.15% |
> | 32 | 41.76% |
> | 64 | 41.53% |
> | 128 | 41.23% |
>
>
> We observe that using more iterations does not improve the performance significantly. The reason may be that traditional diffusion methods aim to use the whole image as the supervision signal to promote generating specific images involving plentiful details. Instead, the goal of our method is to facilitate the generated OOD features to contain content that differs from ID features. In other words, the supervision information about generating OOD features is very simple, resulting in the consequence that using more iterations does not boost the performance.

---

### Meta-Review · Area_Chair_UqUy · 2023-12-06

**Metareview:**

Dear Authors,

Thank you for submitting your draft. Most of the reviews are on the slightly positive side (3,6,6). However, concern remains regarding the contribution, which we believe could be attributed to the writing. We encourage authors to improve readability to make contribution and methodology more clear.

Authors have stated in their rebuttal that "recent researches have shown that the amplitude and phase could be separately regarded as the style and content of the input", however, the work they cite are " Lee et al. (2023); Chen et al. (2021); Oppenheim & Lim (1981)" indicating this information might not be just recent. A simple rephrasing shall clarify this.

We also encourage authors to further clarify in their final copy, how the "diffusion" term being used in their draft is different from the general use of this term nowadays.


Regards

Meta review

**Justification For Why Not Higher Score:**

Rank is only slightly positive. Paper's writing style could be improved. Sitting in Dec. 2023 their comparisons look old.

**Justification For Why Not Lower Score:**

An interesting problem. One reviewer who questioned novelty, still was not interested in decreasing rank, indicating reviewer found the paper somewhat interesting.

---

### Decision · Program_Chairs · 2024-01-16

Accept (poster)